# MW-Net: Multi-Wave U-Net with Cross-Wave Links for Multi-Scale Physical Dynamics

## Abstract

We propose Multi-Wave Network (MW-Net), a novel deep learning architecture for modeling the temporal evolution of complex, multi-scale physical systems. MW-Net extends the U-Net architecture by stacking multiple encoder–decoder "waves" (U-Net modules). Unlike prior stacked U-Net variants such as SineNet, which restrict skip connections to within each wave, MW-Net introduces skip connections both within and across successive waves at matching spatial resolutions. This design enhances hierarchical representation learning by enabling repeated interactions between feature representations at the same and different spatial scales, supporting progressive refinement of learned dynamics and offering explicit control over network depth through the number of stacked waves. We evaluate MW-Net on a diverse set of physical systems: 2D Kolmogorov fluid turbulence, Hasegawa–Wakatani plasma turbulence, a shallow-water planetary atmosphere model, and buoyant smoke flows (2D and 3D). Across all cases, MW-Net consistently outperforms state-of-the-art baselines and achieves Pareto improvements in the accuracy–computational cost trade-off. While the best-performing baseline varied by task, MW-Net achieved substantially lower errors and up to 3× faster convergence in reaching low-error regimes under fixed learning schedules.

## 1 Introduction

Accurate prediction of the temporal evolution of complex, multi-scale physical systems is essential in many domains, including high Reynolds number fluid dynamics (Wilcox et al., 1998), magnetized plasma systems (Hasegawa, 2012), weather forecasting, and atmospheric modeling Lam et al. (2023). The dynamics of these systems are characterized by the emergence of interacting structures across a wide range of spatial scales, manifesting in nonlinear phenomena such as turbulent energy cascades (Kolmogorov, 1962). In such systems, the ratio between the largest and smallest relevant scales can span several orders of magnitude, making high-fidelity numerical simulations computationally expensive or infeasible. Moreover, resolving fast information propagation often necessitates implicit time integration schemes, further increasing computational cost.

This has motivated growing interest in machine learning (ML) approaches for accelerating simulations or building efficient surrogate models. ML can be used in several roles (Wang et al., 2024; Lino et al., 2023): (1) to accelerate traditional solvers by modeling sub-grid physics (Um et al., 2020; Kochkov et al., 2021a; Greif et al., 2023; Belbute-Peres et al., 2020; Beck et al., 2019; Lapeyre et al., 2019; Subel et al., 2021; Obiols-Sales et al., 2020) or learning effective initial conditions (sub, 2025); or (2) to fully replace solvers via learned surrogates (Bhatnagar et al., 2019; Brandstetter et al., 2023; Alkin et al., 2024; Li et al., 2020; Lu et al., 2021; Wang et al., 2021). Surrogate models can be further divided into physics-informed approaches, which incorporate known governing equations into the loss function (Raissi et al., 2019; Karniadakis et al., 2021; Donnelly et al., 2024; Zubov et al., 2021; Zhao et al., 2023; Jin et al., 2021; Eivazi et al., 2022; Li et al., 2024), and purely data-driven models that rely only on observed data (Clavier et al., 2025; Stachenfeld et al., 2022; Wang et al., 2020; Gahr et al., 2024; Lippe et al., 2023; Zhang et al., 2024; Kim et al., 2019).

In this work, we focus on the data-driven setting, motivated by scenarios where the underlying PDE may not be known (e.g., experimental observations) or where low-resolution data may not clearly satisfy the governing equations. While our experiments are conducted on high-fidelity numerical solutions of known PDEs, the data was sampled at relatively large output time steps — a common

practical setup (Stachenfeld et al., 2022) that accelerates surrogate evaluation but leads to violations of the original numerical constraints. This setup emulates aspects of reduced-order modeling and real-world data applications, and supports the development of architectures that do not rely on access to the underlying PDE.

To address these challenges, we introduce Multi-Wave Network (MW-Net), a deep learning architecture designed for efficient modeling of multi-scale physical dynamics. MW-Net builds upon the U-Net architecture by stacking multiple encoder-decoder "waves" (i.e., U-Nets), connected not only within each wave but also across waves via skip connections at matched spatial resolutions This design enables repeated interactions across spatial scales, progressive refinement of learned dynamics, and explicit control over network depth.

**Our main contributions are:**

- We present MW-Net, a new deep learning architecture for modeling multi-scale dynamics in complex physical systems.

- We evaluate MW-Net on four challenging physical systems — 2D Kolmogorov turbulence, Hasegawa–Wakatani plasma turbulence, buoyant smoke flow (2D and 3D), and a 2D shallow-water planetary atmosphere — demonstrating consistent improvements over strong state-of-the-art (SOTA) baselines. MW-Net achieves substantially lower prediction error and 3× faster convergence compared to best-performing baselines.

- We evaluate not only trajectory prediction but also statistical characteristics of the learned dynamics, specifically for the Hasegawa–Wakatani system, which exhibits chaotic behavior with a Lyapunov time shorter than the output sampling interval.

## 2 SURROGATE MODELS FOR MULTI SCALE PHYSICS

Below we summarize major architectural families used for learned surrogates in multi scale physics, highlighting their mechanisms, strengths, and limitations.

### 2.1 FOURIER NEURAL OPERATORS (FNOS)

FNOs (Poli et al., 2022; Tran et al., 2023; Helwig et al., 2023; Li et al., 2020; Rahman et al., 2022) use spectral convolution layers, where each layer: (1) applies a Fast Fourier Transform (FFT) to map the field to the frequency domain; (2) Truncates to a subset of Fourier modes and applies learnable complex-valued weights; and (3) uses an inverse FFT to return to the spatial domain.

**Strengths:** (1) Efficient access to global receptive fields; few spectral modes can capture large scale patterns effectively. (2) Offers a compact basis for smooth fields and long range correlations; complexity per layer scales roughly with the FFT cost ($O(N\log N)$ for N grid points).

**Limitations:** (1) Frequency domain filtering can act as a low pass bias, making sharp local features (thin filaments, sharp fronts) harder to model such that fine scale fidelity may suffer (Liu et al., 2025; George et al., 2022; Roberts, 2025; Guan et al., 2023). (2) Application to non-periodic domains or complex geometries often requires windowing, or alternative bases (Qin et al., 2025). These limitations motivate architectures that preserve nonlocal coupling while explicitly modeling local interactions.

### 2.2 TRANSFORMER-BASED MODELS

Vision style transformers (Vaswani et al., 2017; Dosovitskiy et al., 2020) adapted to 2D/3D physics (McCabe et al., 2024; Cachay et al., 2022; Chattopadhyay et al., 2020; Gao et al., 2022; Nguyen et al., 2023; Alkin et al., 2024) partition the domain into patches and embed each as a token. Self attention exchanges information across all tokens.

**Strengths:** Self-attention naturally captures multi-scale behavior; any patch can attend to any other.

**Limitations:** (1) Attention cost is quadratic in the number of tokens (area in 2D / volume in 3D), which limits scalability. Physics oriented works propose linear/near linear attention (Li et al., 2023b;

Cao, 2021; Hao et al., 2023; Li et al., 2023a), however these lead to a trade off in accuracy comparable to simpler architectures (e.g., FNO) (Li et al., 2023b). (2) Transformers are often data hungry (Abdel-Aty & Gould, 2022). (3) In periodic domains (common in physical modeling), learnable relative (periodic) encodings (Shaw et al., 2018; Wu et al., 2021) add model complexity. These trade-offs make convolutional approaches attractive for their efficiency and inductive biases.

### 2.3 CONVOLUTIONAL MODELS

Convolutional models remain a popular choice for surrogates due to simplicity, inductive biases (translation equivariance), parameter efficiency (weight sharing), and linear scaling with the grid size for fixed kernel sizes.

#### 2.3.1 ENCODE–PROCESS–DECODE (RESNET PROCESSORS)

An encoder downsamples to a latent grid with increased channel count; a processor operates at fixed spatial resolution (often a deep ResNet (He et al., 2016)) to model dynamics; a decoder upsamples back to the original resolution. This Encode–Process–Decode paradigm (Battaglia et al., 2018; Sanchez-Gonzalez et al., 2018; 2020) is frequently used in fluid and plasma surrogates (Cheng & Zhang, 2021; Stachenfeld et al., 2022; Kim et al., 2019).

**Strengths:** Simple, efficient, and effective for local modeling at the processor's resolution. Residual connections ease optimization and permit depth.

**Limitations:** The processor operates at a single resolution; cross scale interactions are learned only indirectly through the encoder/decoder pathway, which can limit fidelity under strong multi scale coupling (e.g., cascades, wave–eddy interactions). To expand receptive fields at constant resolution, dilations are often introduced.

#### 2.3.2 DILATED RESNETS (DILRESNET)

DilResNet replaces/augments standard convolutions with dilated convolutions to enlarge the receptive field without pooling (Stachenfeld et al., 2022).

**Strengths:** Captures long range dependencies while preserving the native grid resolution and fine detail.

**Limitations:** Larger effective kernels substantially increase compute and memory; in practice, DilResNet can require up to an order of magnitude more compute than comparable convolutional models with similar parameter counts (Zhang et al., 2024; Gupta & Brandstetter, 2022; Li et al., 2023b). This motivates multi resolution designs that natively route information across scales.

#### 2.3.3 U-NET AND VARIANTS

A U-Net couples a multi scale encoder and decoder via skip connections at matching resolutions, allowing similar scale features from the encoder to inform the decoder directly.

**Strengths:** Thanks to its strong multi scale inductive bias, computational efficiency, and robust training dynamics, U-Nets (and their many variants) are arguably the most widely adopted baseline for learned surrogates in multi scale physics, often achieving competitive—frequently state of the art—accuracy at a favorable accuracy–cost balance across diverse benchmarks (Zhang et al., 2024; Gupta & Brandstetter, 2022; Ohana et al., 2024). Common variants include Classic U-Net, Attention U-Net, ResUNet (Diakogiannis et al., 2020), and ConvNeXtU-Net (Ohana et al., 2024).

**Limitations and remedies:** (1) U-Nets perform only a single downsampling and upsampling pass, which limits the number of explicit cross-scale interactions during feature processing. (2) As a time-stepping predictor, a U-Net can exhibit temporal misalignment between encoder and decoder embeddings because skip-connected features correspond to different effective times when predicting the next step. SineNets (Zhang et al., 2024) mitigate this by stacking multiple U-Nets sequentially (Xia & Kulis, 2017; Shah et al., 2018), each advancing by a smaller sub-step to reduce misalignment. However, SineNet's skip connections remain confined within each U-Net wave, and information flows across waves only through composition at the highest-resolution level (Zhang et al., 2024).

Attention-augmented variants further strengthen cross-scale interactions but at the cost of higher training complexity and optimization difficulty.

## 3 MODELS

We selected baseline models that have demonstrated strong performance as physics surrogates on multiple systems in prior literature (Zhang et al., 2024; Gupta & Brandstetter, 2022; Ohana et al., 2024). Our focus is on U-Net variants, which consistently outperform other architectures across multiple benchmarks, including the two multi-scale systems evaluated in this paper. For instance, in 2D shallow water and buoyant smoke systems (described in Sections 5.1 and 5.2), U-Net variants outperformed Fourier Neural Operator (FNO) models by nearly an order of magnitude in terms of inverse error-to-computational-cost ratio Zhang et al. (2024). While we do not directly benchmark against FNOs and transformer-based models in this study, our evaluation indirectly reflects their performance through comparative analysis on shared systems.

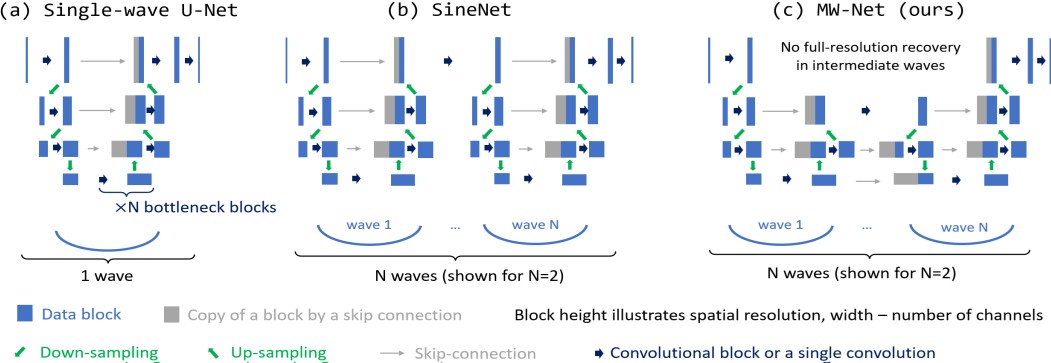

Figure 1: U-Net variants. Detailed description of convolutional blocks is given in Appendix A. The models are illustrated with 4 resolution levels for simplicity (but 5-level variants were used).

### 3.1 U-NET$_{\text{BASE}}$ (BASE VARIANT)

Our primary baseline is the U-Net architecture adapted from Gupta & Brandstetter (2022) (labeled U-Net$_{\text{base}}$). This model closely resembles the original U-Net (Ronneberger et al., 2015) as implemented in PDEBench Takamoto et al. (2022), with minor modifications inspired by modern variants, which include: (1) group normalization (Wu & He, 2018) instead of batch normalization, (2) enabling of bias parameters in convolutional layers, (3) a reduction of bottleneck block at the lowest resolution level to match the parameter count of the original U-Net (corresponding to N=1 in the Fig. 1a).

Each encoder and decoder level contains a convolutional block with two 3x3 convolutions (stride 1, padding 1), a widely adopted standard in convolutional architectures (Zagoruyko & Komodakis, 2016). At the highest-resolution level, the first convolution expands the number of channels to a specified width, while the final convolution restores it to the original count. Downsampling uses 2x2 max pooling, and upsampling uses transposed convolutions.

### 3.2 U-NET$_{\text{MOD}}$ (MODERNIZED VARIANT)

U-Net$_{\text{mod}}$ (also adapted from Gupta & Brandstetter (2022)), incorporates several enhancements from modern U-Net designs Ho et al. (2020); Nichol & Dhariwal (2021); Ramesh et al. (2021):

- Residual skip connections within convolutional blocks, similar to Wide ResNet (Zagoruyko & Komodakis, 2016) and ResUNet-a Diakogiannis et al. (2020).
- Learnable downsampling via $2 \times 2$ convolutions (stride 2) replacing max pooling.
- An enhanced bottleneck block (corresponding to $N = 3$ in the Fig. 1a).

Although some variants include spatial attention blocks Gupta & Brandstetter (2022), we observed no significant performance gains and encountered training instability and increased computational cost. Therefore, attention-based models were excluded from our baseline comparisons.

## 3.3 ConvNextU-Net (CNU-Net)

CNU-Net (adapted from Ohana et al. (2024) where it showed good performance on several physical systems) is another modern U-Net variant that integrates ConvNext blocks (Liu et al., 2022; Xie et al., 2017), which employ the 'divide and conquer' approach to broaden the spatial receptive fields and semantic representations without increasing computational cost. These blocks stack one channel-wise convolution (Chollet, 2017) with a 7×7 filter and two sequential 1×1 pointwise convolutions. The latter expand the channel dimensions by a factor of 4 and contract them back.

## 3.4 SineNet

SineNet (Zhang et al., 2024), Fig. 1b, addresses temporal misalignment in U-Net architectures by stacking multiple U-Nets sequentially, each operating at a reduced effective time step. Notably, each U-Net in the stack has internal skip connections, but no skip connections exist between encoder-decoder pairs across U-Nets. Thereby, deep (low-resolution) features are discarded between waves, limiting semantic continuity. Another feature is using average pooling for downsampling.

## 3.5 MW-Net (ours)

### 3.5.1 Architectural Motivation

In most U-Net variants, the network depth—defined as the total number of convolutional layers—is constrained by two factors. (1) The number of resolution levels is typically fixed at five based on empirical success across physical systems. (2) The number of convolutional layers per level is usually fixed at two. This leaves channel count (i.e., network width) as the primary tunable parameter. However, it is well-established that both depth and width must be carefully balanced to optimize performance. As shown in Bengio & LeCun (2007); Larochelle et al. (2007), and supported by circuit complexity theory, shallow networks may require exponentially more units to match the expressiveness of deeper architectures, which scale polynomially. This motivates architectural designs that allow explicit control over depth, especially for multi-scale physical modeling.

### 3.5.2 Insights from Multi-Scale Physical Dynamics

Features of various scales emerging in complex systems interact predominantly locally. Small-scale features that are spatially separated do not interact directly, but can interact indirectly through larger-scale structures that encompass them. This makes U-Nets well-suited for multi-scale systems: it efficiently creates multi-resolution embeddings and captures local interactions. However, it allows only a single pass of cross-scale interaction, limiting the ability to progressively refine representations through network propagation.

### 3.5.3 Improving Computational Efficiency

In U-Net architectures (including SineNet), the highest-resolution layers—those with the largest spatial dimensions and lowest channel count—are typically the most computationally expensive. At the same time, these layers often encode less semantic information. MW-Net reduces their usage at intermediate waves, substantially improving efficiency without major sacrifices to accuracy.

### 3.5.4 Design and Key Components of MW-Net

MW-Net, Fig. 1c, is constructed by stacking multiple U-Net modules (waves), each with its own encoder–decoder pair. The key innovation lies in introducing **cross-wave skip connections** at matching resolution levels, enabling features to persist and evolve across waves. This facilitates hierarchical learning, repeated multiscale interactions, and progressive refinement of representations.

To improve computational efficiency, MW-Net omits the highest-resolution layers in intermediate waves—these layers are costly yet contribute less semantic information. **Average pooling** is used for downsampling, and **transposed convolutions** for upsampling. Each convolutional block is independently parameterized, avoiding weight sharing and allowing flexible learning.

MW-Net shares structural similarities with **LadderNet** (Zhuang, 2018), originally proposed for medical image segmentation. Both architectures employ intra-wave and cross-wave skip connections. However, MW-Net introduces several key differences:

- **Aggregation method:** LadderNet uses summation; MW-Net uses channel-wise concatenation, preserving feature diversity.

- **Full-resolution recovery**: is avoided in intermediate waves, reducing cost.

- **Skip connectivity:** MW-Net includes skip connections at all resolution levels, including the deepest layers, which LadderNet omits.

- **Skip placement:** Skip connections are placed before concatenation, enhancing gradient flow and feature reuse.

- **Weight sharing:** MW-Net does not share weights across convolutional blocks, allowing more expressive learning.

- **Pooling:** Similar to SineNet, average pooling is used instead of stride-2 convolutions.

### 3.6 U-Net$_{\text{DEEP}}$ (Deeper U-Net)

To isolate the effect of multi-scale interactions, we also implement DeeperU-Net—a single U-Net with deeper convolutional blocks (four layers per block) and internal skip connections. This variant does not use recurrent weight sharing, distinguishing it from R2U-Net (Alom et al., 2018) commonly used in vision applications.

## 4 Basis for Model Comparison

### 4.1 Hyperparameter Selection

To ensure fair comparisons, we standardized hyperparameters across models where possible:

- **Resolution levels**: All models use five levels, consistent with prior work (Gupta & Brandstetter, 2022; Zhang et al., 2024; Ohana et al., 2024).

- **Channel expansion**: A fixed ratio of 2 between resolution levels is used throughout.

- **Activation**: GELU activation (Hendrycks & Gimpel, 2016) is used throughout.

- **Boundary conditions (padding)**: consistently with Gupta & Brandstetter (2022), periodic padding was used for periodic domains; zero padding was used for other boundaries.

### 4.2 Accuracy vs. Cost Trade-off

Rather than comparing best-performing variants alone, we reconstruct the Pareto frontier of accuracy vs. computational cost. Computational cost is defined as training / inference time on a single A100-80GB GPU. Width (channel count) is varied for all models, whereas depth is additionally varied in ablation studies for SineNet and MW-Net.

### 4.3 Training Setup

All models were trained using the ADAM optimizer with a custom learning schedule featuring exponential decay with sinusoidal annealing (see Appendix C). Batch size and epoch count were fixed per problem across models. Learning rate scaling was fine-tuned per model.

## 5 PHYSICAL SYSTEMS AND DATASETS

We evaluate all models on a diverse set of multi-scale physical systems. Here, we provide high-level description of the physical systems, details including data generation and model training strategies are given in Appendix B.

### 5.1 BUOYANT INCOMPRESSIBLE GAS FLOW WITH SMOKE

This system represents thermal convection of light species, e.g., smoke, in a closed rectangular domain. The flow is governed by the incompressible Navier-Stokes equations augmented with a transport equation for the smoke concentration (assuming pure advection). The flow is driven by the buoyancy force which is proportional to the smoke concentration. 2D and 3D systems were modeled. The datasets adopted from Gupta & Brandstetter (2022) and Li et al. (2023b) respectively, were generated using the ΦFlow solver (Holl et al., 2020). In the 2D case, a 128×128 grid was used and the Reynolds number was 100. In the 3D case, a 64×64×64 grid was used and the Reynolds number was 333. A trajectory example is shown in Figs. 9, 10, and 11 in Appendix.

### 5.2 SHALLOW-WATER PLANETARY ATMOSPHERE MODEL

The shallow water (SW) equations are derived by depth-integrating the incompressible Navier-Stokes equations (Vreugdenhil, 2013). One of their applications is for modeling planetary atmospheres, predicting evolution of the pressure field (scalar) and wind velocity field (vector). We adopted the dataset from Gupta & Brandstetter (2022) for a model planet which was generated using a modified SpeedyWeather.jl (Klöwer et al., 2022) solver on a cartesian $192 \times 96$ grid. A trajectory example is shown in Figs. 12, 13, and 14 in Appendix.

### 5.3 KOLMOGOROV FLOW TURBULENCE

The 2D Kolmogorov flow problem is a common benchmark for studying developed fluid turbulence in periodic domains. The sinusoidal flow of viscous liquid is induced by a unidirectional periodic force and the dynamics is governed by the incompressible Navier-Stokes equations. The dataset adopted from Li et al. (2023b) was generated using a modified pseudo-spectral solver with Re = 1000 and forcing factor $f_0 = 8$. The output resolution and time step were 256x256 and 1/16 respectively. A trajectory example is shown in Fig. 15 in Appendix.

### 5.4 HASEGAWA–WAKATANI (HW) PLASMA TURBULENCE

Hasegawa-Wakatani (HW) equations describe turbulence of fully-magnetized plasma in nuclear fusion devices. The model assumes that there a gradient in the plasma density transverse to an external uniform magnetic field. The dynamics is formulated for non-dimensional perturbations of plasma (ion) density $n$ and the electric potential $\phi$. Periodic boundary conditions are used. We have solved these equations for $n$ and $\phi$ using the BOUT++ code (Dudson et al., 2009), for $\alpha = 0.01$ and $\kappa = 0.5$. Spatial resolution was 128x128 and the time step was 1. Trajectory example: Figs. 16 and 17.

## 6 EXPERIMENTS

Following Gupta & Brandstetter (2022); Zhang et al. (2024), all models were trained to predict one future state from a fixed number of past states (concatenated channel-wise), then rolled out autoregressively to generate trajectories (details on the number of time steps are in Appendix B).

**Evaluation protocol.** Following Li et al. (2023b); Zhang et al. (2024), for smoke, shallow-water, and the Kolmogorov flows, we use the *scaled L2* loss computed per time step and averaged across test trajectories:

$$L_{2,\mathrm{t}}(\hat{u}_t, u_t) = \frac{1}{M} \sum_{k=1}^{M} \frac{\|\hat{u}_t^k - u_t^k\|_2}{\|u_t^k\|_2},$$

where $\hat{u}_t$ is the predicted field at time step $t$, $u_t$ is the ground truth field at time step $t$, $M$ is the number of scalar fields, $\hat{u}_t^k$ and $u_t^k$ are the $k$-th scalar fields of the prediction and ground truth, respectively, $\| \cdot \|_2$ denotes the L2 norm over spatial dimensions. Trajectory examples are presented in Appendix E.

For HW turbulence, where the Lyapunov time ($\approx 0.5$ (Pedersen et al., 1996)) is shorter than the output interval ($\Delta t = 1$), trajectory errors accumulate quickly (an example of a trajectory is shown in Figs. 16 and 17 in Appendix); thus, we evaluate statistical fidelity over a single 2000-step rollout using (i) time-averaged spatial FFT spectra and (ii) spatially averaged temporal autocorrelations, an example of which is given in Fig. 7 in Appendix. Here we present aggregated errors for these quantities normalized by ground-truth variance:

$$\text{err}(y) = \frac{\langle (y - y_{\text{true}})^2 \rangle}{\text{var}(y_{\text{true}})},$$

where $y$ denotes the parameter of interest (FFT harmonics or autocorrelation), and $\langle \cdot \rangle$ denotes the average over the x-axis in Fig. 7 ($k$ for spectra, time for autocorrelations).

**Training details.** Loss: scaled L2 loss was used for smoke, SW, Kolmogorov's flow; MSE for HW turbulence. Each model was trained with 3 initializations (6 for HW and Kolmogorov's flow) using fixed seeds $\{2, 12, 22, \dots\}$; we report the best-performing variant over the initializations.

## 6.1 PARETO TRADE-OFF BETWEEN ACCURACY AND COST

We vary model width starting from 4 channels at the highest-resolution level, doubling for each bigger model. The upper limit on the model width was dictated by the training budgets capped at 8 h (smoke, shallow-water), 5 h (Kolmogorov), and 2000 s (HW) wall clock time, all on a single A100–80GB GPU.

We benchmark the single-wave U-Net variants against SineNet (2 waves) and MW-Net (3 waves); SineNet was used with fewer waves due to higher computational cost. Limited 3D smoke experiments were run, with U-Net$_{\text{base}}$ being the only baseline.

**Smoke, Kolmogorov's flow, and Shallow-Water model:**

Figure 2a-d plots final-step relative $L_2$ error vs. training time for smoke, shallow-water, and Kolmogorov's turbulence. Error vs. inference-time trends are presented in Fig. 5 in Appendix. A clear Pareto frontier emerges: wider models improve accuracy but increase cost. Error drops steeply for narrow widths and saturates for wider ones.

**MW-Net-3** consistently outperforms all baselines, reaching lower errors faster and maintaining its advantage as cost grows. The performance of base-lines is case-dependent with no clear winner. Notably, **U-Net$_{\text{deep}}$** performs best on 2D smoke and second-best on shallow-water and Kolmogorov, indicating that the model depth is important for accuracy but a single encoder–decoder pass is insufficient.

Relative to the best non-MW baseline, **MW-Net-3** reduces error by $\sim$30% (shallow-water), $\sim$20% (Kolmogorov), and $\sim$10% (smoke), translating to **2–3$\times$** less training time to reach equivalent accuracy. These gains persist across model rollouts, starting from the first steps (first-step errors are reported in Fig. 5).

**Cross-family comparison for Kolmogorov's flow (using literature data):**

We also overlay published per-step rollout errors for the Kolmogorov flow by non-U-Net models (FactFormer, DilResNet, FNO, F-FNO, see Fig. 2f). Notably, U-Net-based models—especially **MW-Net-3**—achieve substantially lower errors at similar or shorter inference times ($< 1$ s for U-Net-based models vs. $\sim$5 s for DilResNet vs. $\sim$1 s for other literature models). While optimization choices (e.g., batch size) may affect absolute values, they cannot account for the observed order-of-magnitude accuracy gap.

**HW turbulence (statistical fidelity):**

Figure 2e reports normalized spatial FFT errors for $n$. Spatial FFT errors for $\phi$ and temporal autocorrelation errors for both fields are presented in Fig. 6. **MW-Net-3** outperforms other models by

large margins, with almost an order of magnitude improvement for the FFT of $n$ and more than an order-of-magnitude improvement for $\phi$. Increasing width does not always help for the HW setup; most models degrade at large widths.

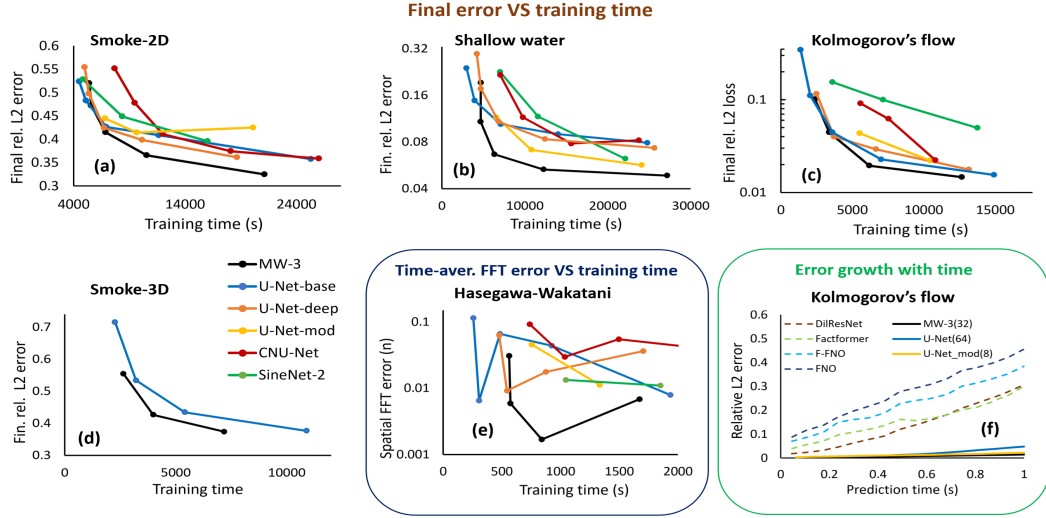

Figure 2: Accuracy vs. training time for all systems + accuracy vs. time step in a roll-out for Kolmogorov's flow.

## 6.2 ABLATION ON THE NUMBER OF WAVES

We conducted ablation studies on the smoke and shallow-water systems to assess the effect of stacking additional U-Net waves. Results in Figure 3 show no noticeable improvement beyond two waves for either MW-Net or SineNet. This suggests that, for these systems, two waves are sufficient to capture the relevant multi-scale dynamics.

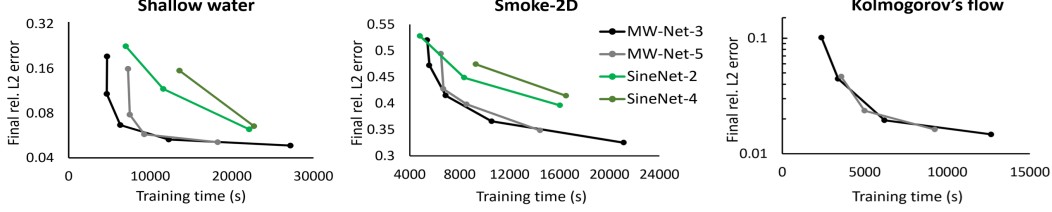

Figure 3: Errors for MW and SineNet models with various numbers of waves

## 7 CONCLUSION

We introduced MW-Net, a novel deep learning architecture designed for modeling multi-scale physical dynamics through stacked U-Net modules with cross-wave skip connections. MW-Net enables deeper and more efficient representation learning by facilitating repeated interactions across spatial scales while reducing reliance on high-resolution layers. Across a range of physical systems, including turbulent and transient regimes, MW-Net consistently outperformed strong baselines, achieving lower prediction errors and faster convergence. In particular, MW-Net demonstrated Pareto improvements in the accuracy–cost trade-off, with up to 3× reductions in training time to reach baseline-level accuracy and multifold improvements in statistical fidelity for chaotic systems. These results highlight MW-Net's potential as a robust and scalable surrogate model for complex physical simulations.

## 8 REPRODUCIBILITY

We provide the implementation of the experiments in the form of an anonymized OSF repository, available at:

`https://osf.io/ch6da/?view_only=8200f9eeeba743a694f4b1a707ed101c`

Please navigate to the 'Files' tab at the top of the screen to access the contents.

The repository includes: instructions in readme.txt, dependencies listed in requirements.txt, code for training models and analyzing results, and the dataset for Hasegawa-Wakatani turbulence.

Each experiment was conducted on a single A100-80GB GPU. A detailed description of the models, datasets, and training procedure can be found in Sections 3-6 of the paper, as well as in Appendices A-C.

## 9 LIMITATIONS

This work primarily focuses on autoregressive prediction tasks using high-fidelity simulated data. Extending MW-Net to settings involving noisy or experimental data and data assimilation remains an open direction for future research. While limited 3D tests indicate promising scalability, broader evaluation across high-dimensional systems is needed. Additionally, MW-Net has so far been applied only to regular Cartesian grids; adapting the architecture to unstructured meshes via graph-based convolutions (Pfaff et al., 2020; Kurz et al., 2025; Gruber et al., 2022; Grigorev et al., 2023; Fortunato et al., 2022), as demonstrated in U-Net GNNs (Gladstone et al., 2024; Deshpande et al., 2024), could enable broader applicability to complex geometries. Finally, although MW-Net shares structural similarities with architectures used in computer vision, its potential for tasks such as semantic segmentation has not yet been explored.

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

## A    U-NET VARIANTS

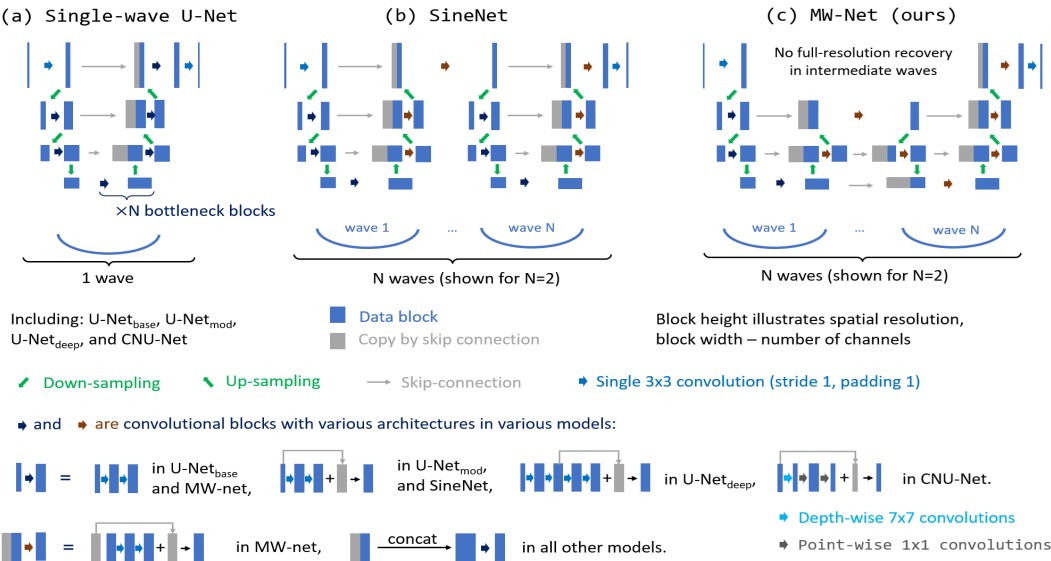

Figure 4: U-Net variants with details of the convolutional blocks.

## B    DETAILS OF THE SYSTEMS, DATASETS, AND MODEL TRAINING

### B.1    THE KOLMOGOROV FLOW

The two-dimensional Kolmogorov flow problem is a benchmark for studying developed fluid turbulence in periodic domains. The sinusoidal flow of viscous liquid is induced by a unidirectional periodic force. The dynamics is governed by the incompressible Navier-Stokes equations in the vorticity form (non-dimensional):

$$\frac{\partial \omega(x,t)}{\partial t} + u(x,t) \cdot \nabla \omega(x,t) = \frac{1}{\text{Re}} \nabla^2 \omega(x,t) + f(x), \qquad x \in (0, 2\pi)^2, \ t \in (0, T],$$

$$\nabla \cdot u(x,t) = 0, \qquad x \in (0, 2\pi)^2, \ t \in [0, T],$$

$$\omega(x,0) = \omega_0(x), \qquad x \in (0, 2\pi)^2.$$

Here, $x$ and $y$ are spatial coordinates, $u$ is velocity (directed along $y$), $\omega$ is vorticity, $Re$ is the Reynolds number and $f$ represents the driving force along the y-direction (Kochkov et al., 2021b):

$$f(x) = -f_0 \cos(f_0 x) - 0.1 \, \omega.$$

The dataset adopted from Li et al. (2023b) was generated using a modified pseudo-spectral solver (Zheng, 2023), for $Re = 1000$ and with forcing factor $f_0 = 8$. The dataset comprises 100 trajectories for training and 20 trajectories for testing, where each trajectory has 160 states. Initial condition $\omega_0$ was sampled from a Gaussian random field following Li et al. (2024), ensuring a broad range of spatial scales in the flow.

The models were trained for 32 epochs to predict 1 time step ahead taking a single time step as input. The batch size was 20. Six initializations with fixed seeds were used for each model. The

models were then rolled-out auto-regressively to generate trajectories of 16 time steps on a batch of 10 trajectories with randomly selected starting time frames. Relative L2 loss was used for both training and presenting the results.

The following learning rate scaling factors were used for the models: U-Net$_{base}$ – 0.25, U-Net$_{mod}$ – 0.125, CNU-Net – 0.25, U-Net$_{deep}$ – 1, MW-Net – 0.25, SineNet – 0.25.

## B.2 BUOYANT INCOMPRESSIBLE GAS FLOW WITH SMOKE (2D AND 3D)

This system represents thermal convection of light species, e.g., smoke, in a closed domain. The flow is governed by the incompressible Navier-Stokes equations which assume that the flow velocity is too low to affect fluid density (Mach number $<< 1$, which is true for thermal convection). The equations are augmented by a transport equation for smoke concentration (assuming pure advection) and are solved in non-dimensional form:

$$\frac{\partial u(x,t)}{\partial t} + u(x,t) \cdot \nabla u(x,t) = \frac{1}{\text{Re}} \nabla^2 u(x,t) - \nabla p(x,t) + f(x,t), \quad x \in (0,L)^n, \ t \in (0,T],$$

$$\frac{\partial d(x,t)}{\partial t} + u(x,t) \cdot \nabla d(x,t) = 0, \qquad\qquad x \in (0,L)^n, \ t \in (0,T],$$

$$\nabla \cdot u(x,t) = 0, \qquad\qquad x \in (0,L)^n, \ t \in [0,T],$$

$$u(x,0) = 0, \quad d(x,0) = d_0(x), \qquad\qquad x \in (0,L)^n.$$

Here, $u$ is the velocity vector, $p$ is the pressure, $d$ is the concentration of light-weight species (e.g., smoke), $f$ is the vertically-directed buoyancy force, which is proportional to the smoke concentration $d$ with a factor 0.5. $n$ denotes the number of spatial dimensions.

Dirichlet boundary conditions are applied to the velocity, and Neumann conditions to the smoke concentration.

### B.2.1 2D CASE

We use the dataset from Gupta & Brandstetter (2022), generated using the ΦFlow solver (Holl et al., 2020) on a 128×128 grid with an output time step of 1.5. The domain size is 32×32, and the Reynolds number is 100. The dataset contains 5,200 training trajectories and 1,300 test trajectories, each with 14 time steps from randomly sampled initial conditions.

Following Gupta & Brandstetter (2022), the models are trained to predict one time step ahead using the previous four time steps (concatenated channel-wise) as input. Models are then rolled out autoregressively to predict time steps 5–14. Each model is trained with 3 fixed-seed initializations (the best-performing realization is used). The error metric was the scaled L2 loss computed per time step, used for both training and evaluation. The models were trained for 80 epochs with batches of 40 time steps. A batch of 30 test trajectories was randomly selected for evaluation. The errors for the first and the last time step are presented, averaged across the test trajectories.

The following learning rate scaling factors were used for the models: U-Net$_{base}$ – 1, U-Net$_{mod}$ – 0.25, CNU-Net – 1, U-Net$_{deep}$ – 1, MW-Net – 1, SineNet – 0.125.

### B.2.2 3D CASE

To assess model scalability, we include a 3D version of the smoke flow system. The dataset, adopted from Li et al. (2023b), was also generated using the ΦFlow solver on a 64×64×64 grid with a time step of 0.75 and Reynolds number of 333. The relative L2 loss was also used for training and evaluation.

The dataset consists of 2,000 training trajectories and 200 test trajectories, each with 20 time steps. Models were trained for 10 epochs with a batch size of 20. 3D convolutions use the same filter

sizes and channel expansion ratio (i.e., ×2) as in 2D. Performance is compared against the U-Net$_{\text{base}}$ baseline.

A learning rate scaling factor of 1 was used for both models.

## B.3 Shallow-Water Planetary Atmosphere Model

The shallow water (SW) equations are derived by depth-integrating the incompressible Navier-Stokes equations (Vreugdenhil, 2013). One of their applications is for modeling planetary atmospheres, predicting evolution of the pressure field (scalar) and wind velocity field (vector). We adopted the dataset from Gupta & Brandstetter (2022) for a model planet, generated using a modified SpeedyWeather.jl (Klöwer et al., 2022) solver. A cartesian grid $192 \times 96$ was used in combination with a fixed output time step of 48 h.

The training and test data consisted of 5,600 and 1,400 trajectories respectively, each trajectory having 11 time steps. The models were trained for 80 epochs to predict one time step ahead (for time steps 3-11) using two previous time steps as input. The batch size was 36. Three initializations with fixed seeds were used for each model. The model was then run autoregressively to predict 9 time steps (3 to 11). A batch of 30 trajectories were randomly selected from the test set to produce the presented results.

The following learning rate scaling factors were used for the models: U-Net$_{\text{base}}$ − 0.25, U-Net$_{\text{mod}}$ − 0.25, CNU-Net − 0.25, U-Net$_{\text{deep}}$ − 1, MW-Net − 1, SineNet − 0.25.

## B.4 Hasegawa-Wakatani Plasma Turbulence

The Hasegawa-Wakatani (HW) equations Hasegawa & Wakatani (1983) describe turbulence relevant to fully-magnetized plasma in nuclear fusion devices. The model assumes a gradient in plasma density transverse to an external uniform magnetic field. The equations are formulated for normalized (non-dimensional) perturbations of plasma (ion) density $n$ and electric potential $\phi$ ($n$ is normalized to the background plasma density, and $\phi$ is normalized to the electron temperature):

$$\frac{\partial n}{\partial t} + \{\phi, n\} + \kappa \frac{\partial \phi}{\partial y} = \alpha(\phi - n) - D_n \nabla^4 n,$$

$$\frac{\partial}{\partial t} \Delta \phi + \{\phi, \Delta \phi\} = \alpha(\phi - n) - D_p \nabla^4 \phi.$$

Here, $x$ and $y$ are the spatial coordinates (the background density gradient is in the $x$ direction). $\kappa$ and $\alpha$ are non-dimensional parameters representing the density gradient and plasma adiabaticity. $D_n$ and $D_p$ are hyper-diffusivity parameters added for numerical stability. The Poisson bracket in the HW equations is defined as:

$$\{A, B\} = \frac{\partial A}{\partial x} \frac{\partial B}{\partial y} - \frac{\partial A}{\partial y} \frac{\partial B}{\partial x}.$$

Periodic boundary conditions are used.

We solve these equations for $n$ and $\phi$ using the BOUT++ code (Dudson et al., 2009), for $\alpha = 0.01$ and $\kappa = 0.5$. The hyper-diffusivity parameters were set to small values, $D_n = D_p = 0.0001$, to ensure numerical stability without affecting the results. Computations were performed on a high-performance computing cluster utilizing eight A100 GPUs. Spatial resolution was 128x128 with a time step of 1. The solver completed the task in approximately 3 hours. A single trajectory was modeled, initiated from white noise. The first 500 time steps corresponded to the warm-up stage, followed by instability growth and saturation. The subsequent 4,300 time steps corresponded to developed (quasi-steady) turbulence. Of those time steps, 4,000 were used for training and 300 for testing.

The models were trained to predict 1 time step ahead using 1 time step as an input. Batch size was 40, with 160 training epochs. The models were then rolled-out auto-regressively to generate

trajectories of 2000 time steps. Since the Lyapunov time for HW turbulence is about 0.5 (Pedersen et al., 1996), i.e., smaller than the output timestep, we compare statistical characteristics of the generated turbulence to the ground truth (no tracing of individual trajectories).

## C  LEARNING SCHEDULE

A custom learning rate $lr$ schedule was applied for all models, based on a warm-up stage followed by an exponential decay combined with cosine annealing Loshchilov & Hutter (2017), as determined by the following expression:

$$lr = 0.01 \cdot \alpha \cdot \exp\left(-5\frac{\max(i, N_{\text{warm}}) - N_{\text{warm}}}{N_{\text{total}}}\right) \cdot \left(0.8 + 0.5 \cdot \sin\left(2\pi\left(0.75 + \frac{i}{N_{\text{warm}}}\right)\right)\right).$$

Here, $i$ is the epoch number, $N_{\text{total}}$ is the total number of epochs reserved for learning, $N_{\text{warm}} = N_{\text{total}}/2$ corresponds to the linear warm-up stage, $\alpha$ is the scaling factor, which was fine-tuned in the range 0.125 - 1.0 for each model.

## D  ACCURACY VS. COST TRADE-OFF (ADDITIONAL RESULTS)

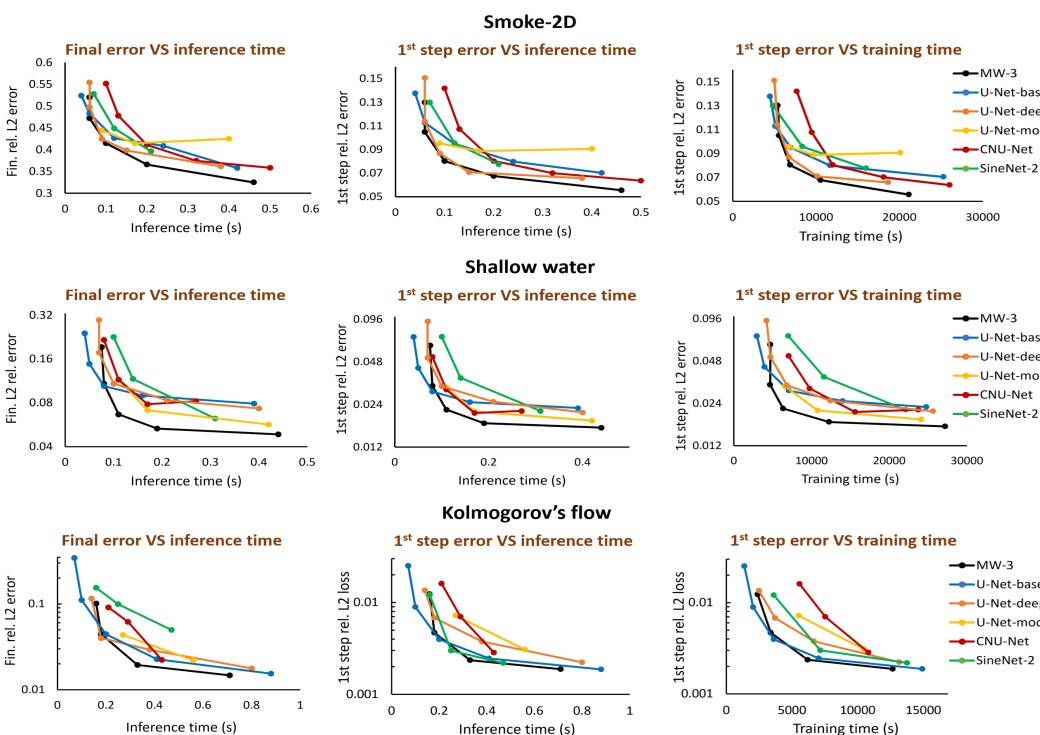

Figure 5: Additional results for the 2D systems where individual trajectories were traced.

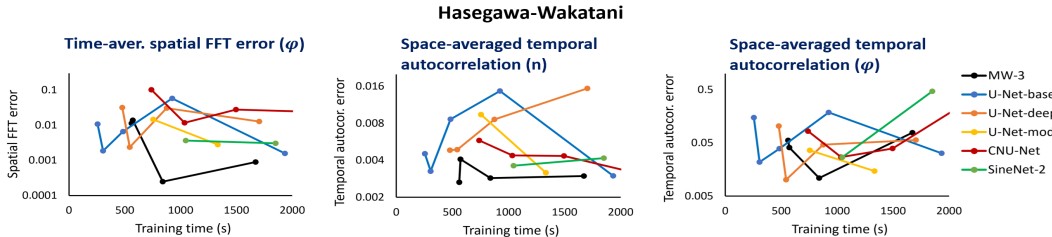

Figure 6: Additional results for the HW system. Examples of the FFT spectra and temporal auto-correlations for which aggregated errors are shown here are given in Fig. 7.

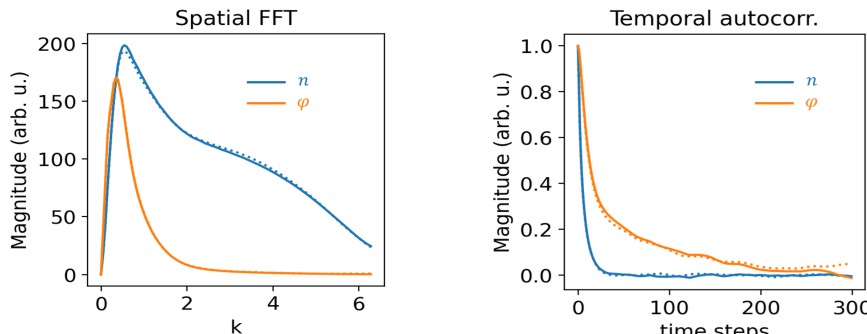

Figure 7: Time-averaged spatial FFT spectra for $n$ and $\phi$ (left) and spatially-averaged temporal auto-correlation for $n$ and $\phi$ (right). Solid lines – ground truth (simulation data), dotted lines – results of the best MW-3 model.

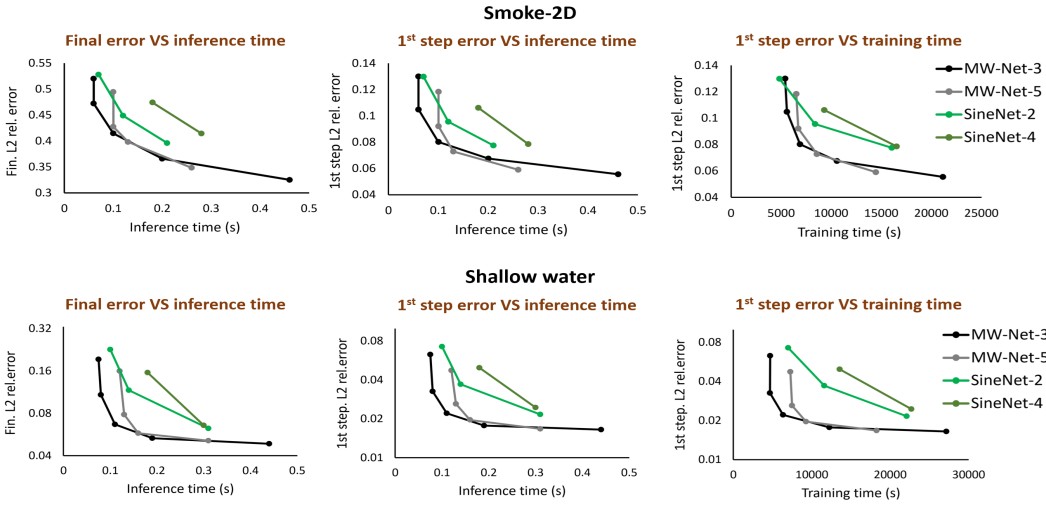

Figure 8: Additional comparison results of MW-Net and SineNet models with more waves on two systems.

# E    MODEL ROLLOUTS

## E.1    BUOYANT FLOW OF SMOKE

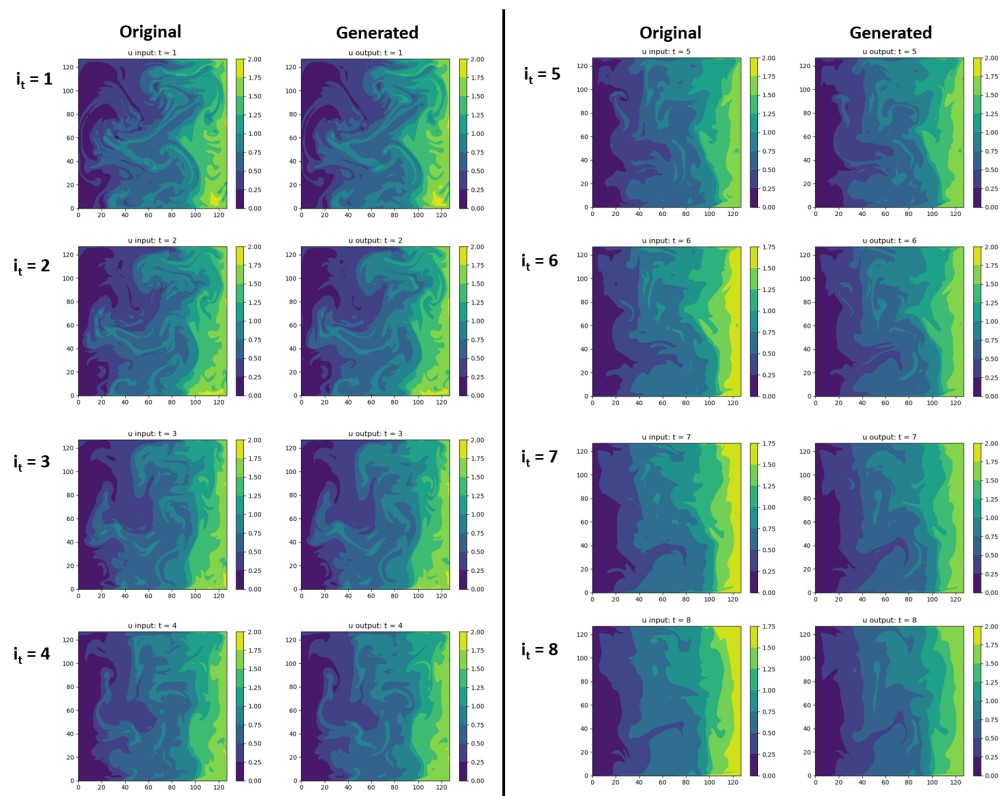

Figure 9: Example of a trajectory for the buoyant smoke flow generated by the MW-Net-3 model (best realization). The field of smoke density.

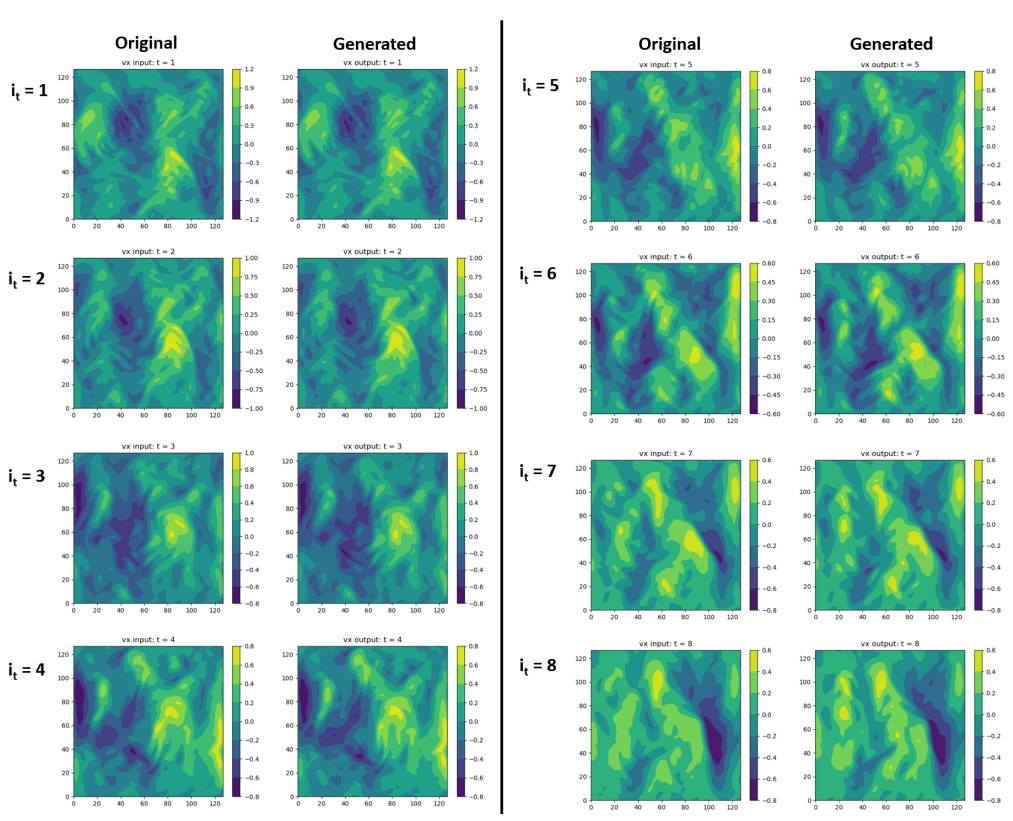

Figure 10: Example of a trajectory for the buoyant smoke flow generated by the MW-Net-3 model (best realization). The field of velocity (x-component).

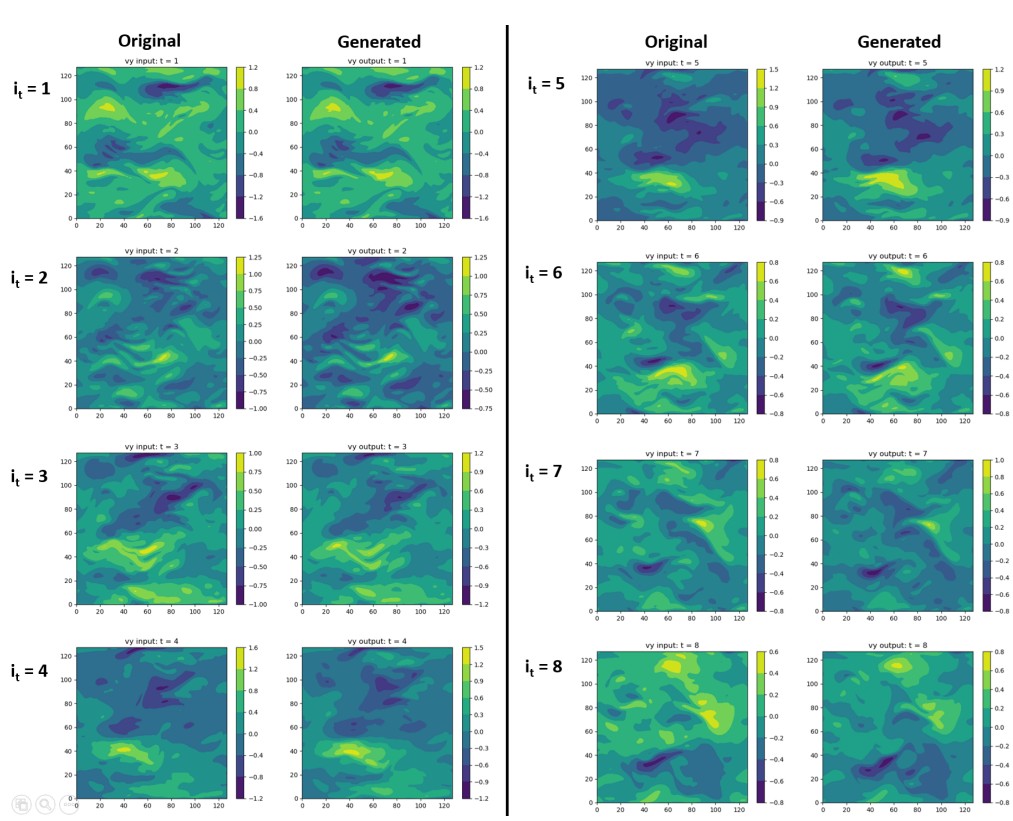

Figure 11: Example of a trajectory for the buoyant smoke flow generated by the MW-Net-3 model (best realization). The field of velocity (y-component).

## E.2 THE SHALLOW WATER SYSTEM

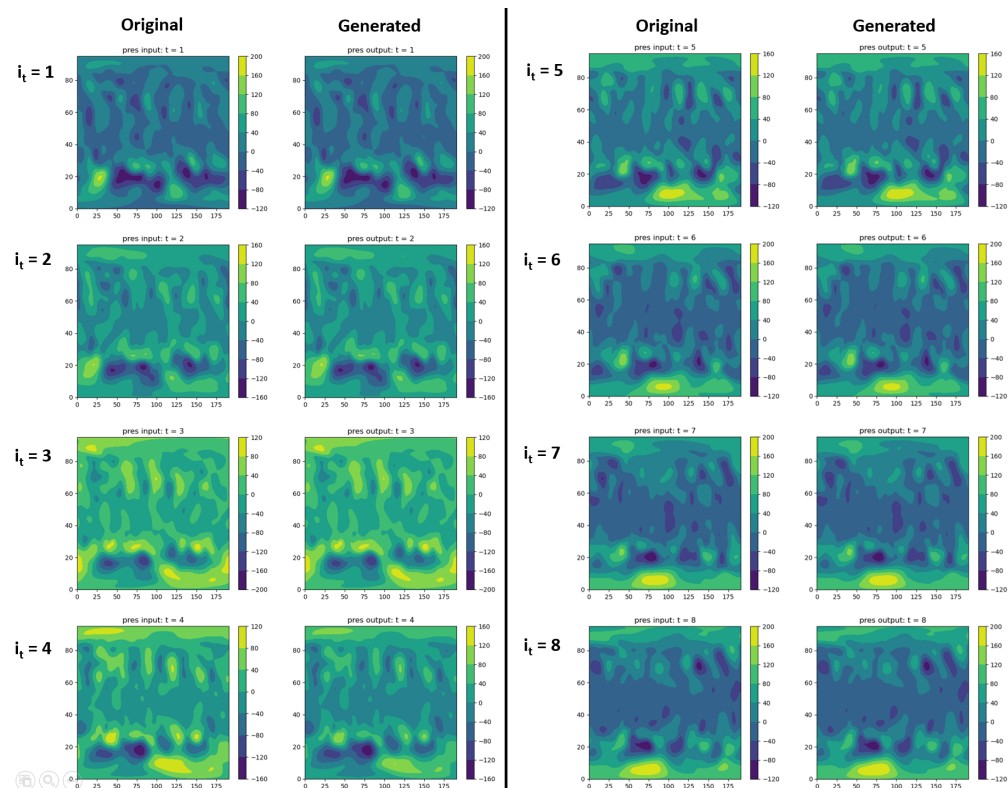

Figure 12: Example of a trajectory for the shallow water system generated by the MW-Net-3 model (best realization). The field of pressure.

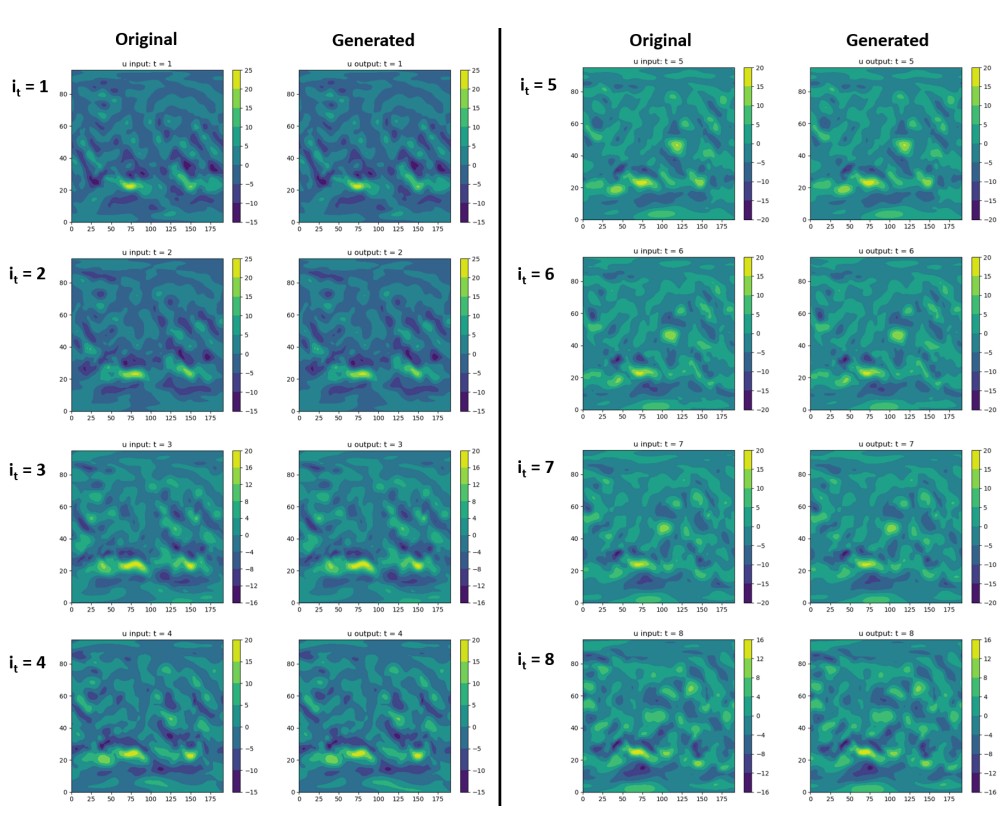

Figure 13: Example of a trajectory for the shallow water system generated by the MW-Net-3 model (best realization). The field of velocity (x-component).

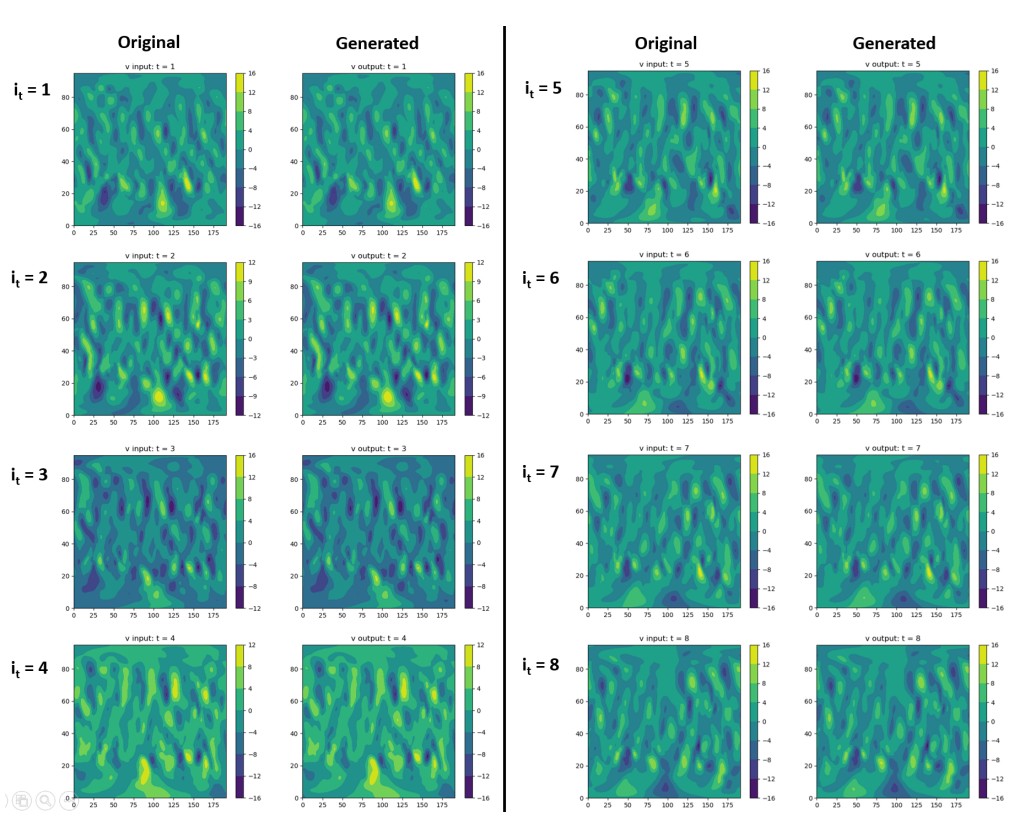

Figure 14: Example of a trajectory for the shallow water system generated by the MW-Net-3 model (best realization). The field of velocity (y-component).

### E.3 KOLMOGOROV'S FLOW

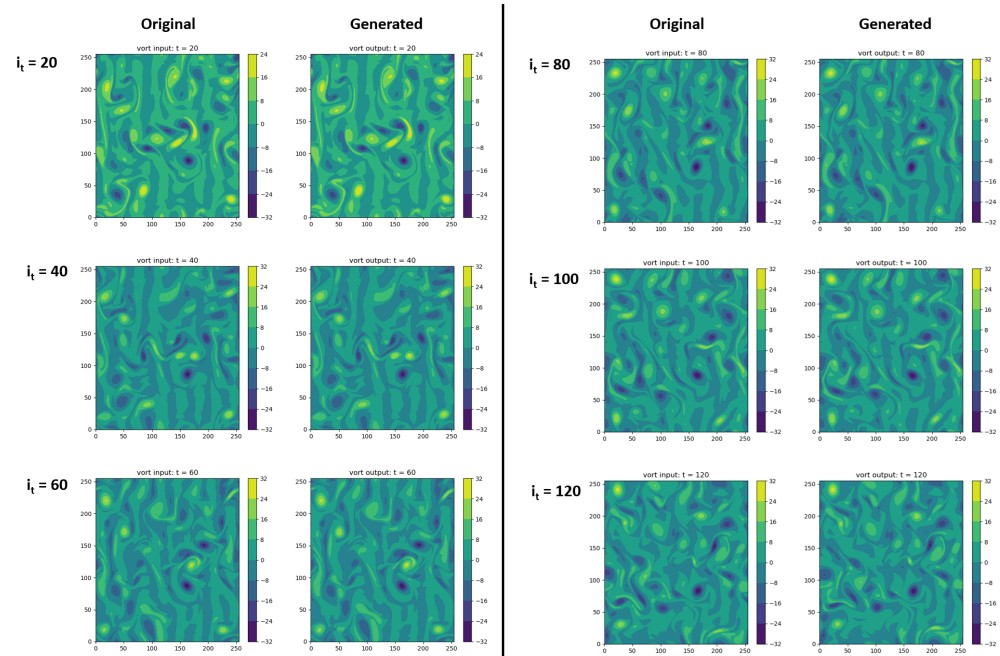

Figure 15: Example of a trajectory for the Kolmogorov turbulence generated by the MW-Net-3 model (best realization). The field of vorticity. Good agreement persists until the end of the trajectory of 120 time steps.

## E.4    HASEGAWA-WAKATANI TURBULENCE

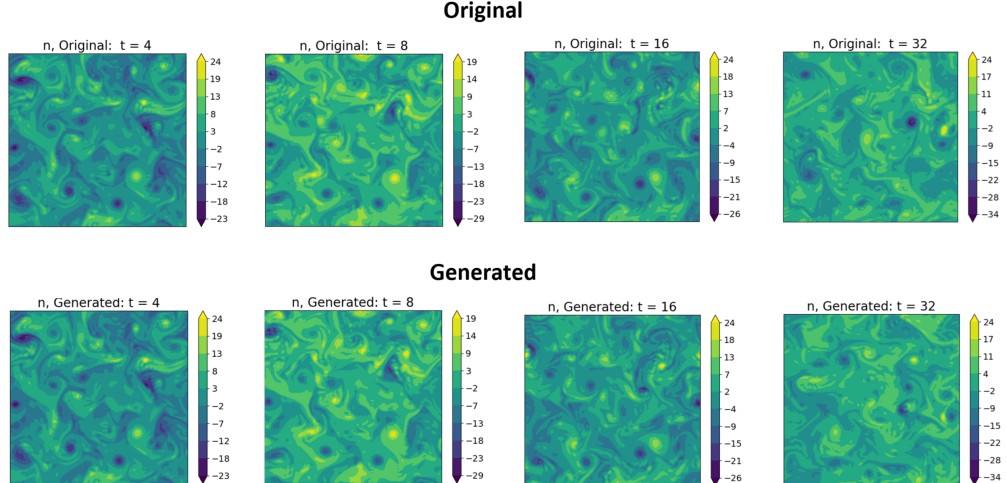

Figure 16: Beginning of a 2000 time step trajectory for HW turbulence generated by the MW-Net-3 model (best realization). The field of $n$: Generated data - bottom row vs. numerical simulation (using BOUT++) - top row. At first, the generated solution resembles the original (simulated) one quite closely. However, with time, the differences amplify and by the time step 32 become significant.

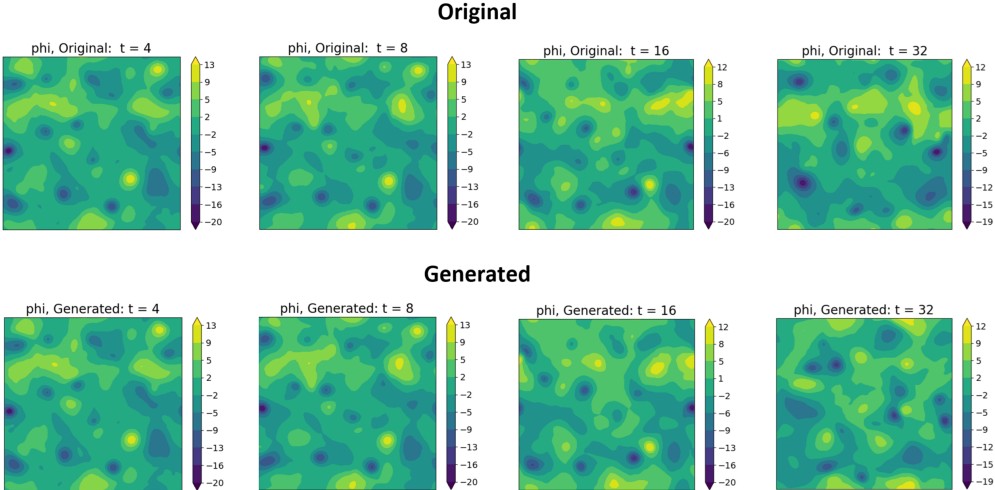

Figure 17: Beginning of a 2000 time step trajectory for HW turbulence generated by the MW-Net-3 model (best realization). The field of $phi$: Generated data - bottom row vs. numerical simulation (using BOUT++) - top row. At first, the generated solution resembles the original (simulated) one quite closely. However, with time, the differences amplify and by the time step 32 become significant.

