# OpenReview forum: "MW-Net: Multi-Wave U-Net with Cross-Wave Links for Multi-Scale Physical Dynamics"
_ICLR.cc/2026/Conference — Submitted to ICLR 2026_

### Official Review · Reviewer_NzpB · 2025-10-27

**Soundness:** 2
**Presentation:** 2
**Contribution:** 2
**Rating:** 4
**Confidence:** 2

**Summary:**

Introducing Multi-Wave Network (MW-Net), a novel architecture designed to model temporal evolution of complex, multi-scale physical systems. In MW-Net multiple U-Net modules, referred to as "waves," are stacked, and skip connections operate both within each wave and, across successive waves at matching spatial resolutions. This cross-wave connectivity is designed for progressive refinement of feature representations through repeated interactions across different spatial scales. The authors evaluate MW-Net on a diverse set of physical systems, demonstrating consistent strong performance and computational efficiency.

**Strengths:**

- **Originality**: the architectural innovation--stacking U-Nets, waves, and adding cross-wave skip connectionsm, addressing limitations of prior stacked U-Nets elegantly. This design allows for progressive refinement of multi-scale features, which is a well-motivated property for physical systems.

- **Quality**: the quality of the paper is in its technical soundness and evaluations -- the authors provide details regarding all design choices, benchmarks and baseline approaches and evaluations of several key properties including both train time and performance accuracy. Further ablations as well as anonymized code are provided, increasing the utility of the framework.

- **Clarity**: the paper is relatively well-written and esy to follow. The motivation for the architecture is given by thoroughly reviewing the limitations of existing models. The architectural differences between MW-Net and its predecessors are explained clearly and accompanied by a visual representation. At last, the experimental setup and results are presented in a transparent and easy-to-interpret manner.

- **Significance**: the significance of the work stems from the wide usage of U-Nets for scientific applications. By providing a new baseline  for data-driven modeling of multi-scale physical dynamics, it presents an opportunity for novel discovery.

**Weaknesses:**

- **Missing benchmarks / limited novelty**: the authors properly address the similarity/differences to LadderNet, however performance of the latter is not reported. Given the close conceptual overlap, a direct comparison would strengthen the claims for novelty and contribution of this work.

- **paper organization**: while the lengthy introduction, background and details are much appreciated the balance between these and contribution of the paper could be improved, putting more emphasis on the presented method and its evaluation (the method itself is only discussed formally in pg. 5 and experimental results only at the end of pg. 7)

- **physical relevance / insight**: the paper demonstrates that MW-Net work better in practice but provides limited insight into _why_ the cross-wave connections are effective. There is no confirmation for the motivation/hypothesis that this enable "progressive refinement" and "repeated multiscale interactions,".

**Questions:**

Following the weaknesses above:
1. LadderNet baseline; is it possible to add this evaluation?
2. physical relevance; is it possible to provide some deeper understanding into the sources of success linking it to physical properties of the systemd.
3. paper organization; it will be valuable to frame the paper more focused on the contribution of this work.

---

### Official Review · Reviewer_bhG9 · 2025-10-29

**Soundness:** 2
**Presentation:** 1
**Contribution:** 2
**Rating:** 2
**Confidence:** 4

**Summary:**

The authors propose MW-Net, a U-Net–style surrogate for PDE dynamics that stacks multiple U-Nets and adds skip connections across U-Nets at matching resolutions so features can persist and be refined over several passes. Compared to prior stacked U-Nets (SineNet), MW-Net also omits the full high-resolution stage in intermediate U-Nets to cut compute while keeping low-/mid-res processing deep. Experiments on 2D smoke, shallow-water atmosphere, Kolmogorov turbulence show lower errors and faster convergence than SineNet, with reported up to 3x less training time for a given accuracy.

**Strengths:**

1. The paper clearly argues why single-pass U-Nets underuse cross-scale interactions and motivates stacking U-Nets with cross-U-Nets skips so features can persist and refine across scales.
2. Omitting the expensive full-resolution layers in intermediate waves cuts compute without big accuracy loss.

**Weaknesses:**

1. The introduction and related-work sections spend many pages on textbook background and prior-work summaries, leaving comparatively little space for the actual method and empirical analysis. Please condense the background and move tutorial material to an appendix.
2. Presentation quality needs a major revision. The paper often reads like a course report rather than a conference manuscript. Replace colon-separated fragments in Section 6 and line breaks with complete paragraphs and topic sentences; use LaTeX structure (e.g., \paragraph{}) instead of ad-hoc lists.
3. The novelty appears to be (a) cross-wave skip connections at matched scales and (b) dropping the highest-resolution stage in intermediate waves. These are sensible engineering choices, but they seem like **incremental variations** on prior stacked/ladder U-Net designs.
4. It’s hard to gauge practical benefit at a glance from the figures. Please add summary tables that, for each dataset, report: metric(s), improvement over the strongest baseline to target accuracy and inference latency/throughput on a specified GPU.

**Questions:**

See weaknesses.

---

### Official Review · Reviewer_zddj · 2025-10-30

**Soundness:** 2
**Presentation:** 2
**Contribution:** 1
**Rating:** 2
**Confidence:** 3

**Summary:**

The authors propose multi-wave networks, an architectural modification to U-Nets to enable better representation mixing across various spatial scales. The authors demonstrate the benefits of their architectural modifications compared to previously proposed methods on a diverse set of dynamical systems.

**Strengths:**

- The proposed modifications enable higher number of skip connections and encourage higher interactions across different spatial resolutions.
- The new architecture shows faster convergence compared to other UNet based architectures like SineNet.

**Weaknesses:**

- The modifications proposed are quite limited in novelty and there is no ablations to understand if the improvements are truly from higher number of interactions across various spatial scales.
- The comparison is limited to convolution based architectures.
- The experiments are quite limited.
- The computational cost (defined as training / inference time on a single A10080GB GPU) is mentioned but there are no tables to understand the quantitative improvements.
- This paper at it's current stage feels like a work in progress and needs more thorough experiments and discussions to be above the ICLR bar of acceptance.

**Questions:**

- How does the learning change with increased depth and width?
- Does the architecture adapt to training - inference resolution mismatch?

---

### Official Review · Reviewer_r5bK · 2025-10-31

**Soundness:** 4
**Presentation:** 4
**Contribution:** 2
**Rating:** 4
**Confidence:** 4

**Summary:**

The authors propose a U-net variant in the form of Multi-Wave-Net (MW-Net), a deep learning architecture for modeling multi-scale physical dynamics. MW-Net extends the traditional U-Net by stacking multiple encoder–decoder “waves” with cross-wave skip connections at matching spatial resolutions. They show how this architectural choice improves hierarchical representation and efficiency by allowing for feature interaction repeatedly across scales. The authors evaluate MW-Net on four systems, 2D and 3D buoyant smoke flow, 2D Kolmogorov turbulence, a shallow-water planetary atmosphere model, and Hasegawa–Wakatani plasma turbulence. Across all cases, MW-Net performs quite well, achieving both 10 to 30% lower prediction errors and fast convergence. Ablation studies show that two stacked waves suffice for good performance.

**Strengths:**

1. The validation of the method across several types of systems is good to see, particularly in the case of turbulence. This bolsters the case of the model's robustness.

2. The authors nicely link their architectural design choices to physical reasoning and then support those links with controlled ablations on network depth and number of waves. It's an elegant approach, but it should be taken further (see weaknesses).

3. I liked the Pareto-front comparisons of accuracy versus cost, which showed consistent quantitative improvements and offered a transparent view of the performance–efficiency trade-off.

**Weaknesses:**

As the authors admit, MW-Net is very similar to 2018's LadderNet, though with some "key" differences--key's scare quotes indicating that, in my view, the architectural changes feel incremental. This tension is heightened by LadderNet's use of cross-wave skip connections, which the authors claim as their own "key" feature. They also don't benchmark against this competing model. Seeing actual scores would help me believe that those architectural changes are "key". If it's not possible to directly compare to laddernet, then could you try altering a few of those 6 points of difference with laddernet to see if your results change? E.g. what happens if you turn on weight sharing or change how pooling works to be more ladder-like?

Overall, the paper reads like a very competent, numerically performative but incremental development. A few more ablation results would help me understand why it is not. An explicit comparison to laddernet would be best, though I understand that's not always feasible.

**Questions:**

Synthesizing from my earlier remarks:

Can you show us in a straightforward, explicit way why those changes from LadderNet make the difference? More generally, can you argue against the work being a minimal, incremental change from previous approaches? Your numerical results are sound and speak for themselves, so this is not a dire issue, but it would really solidify the paper.

---

### Meta-Review · Area_Chair_4jcj · 2026-01-06

**Summary:**

All reviewers are negative on this paper and no author rebuttals were provided.

**Reviewer Concerns:**

No author rebuttals were provided.

**Reviewer Scores:**

All reviewers are negative on this paper.

---

### Decision · Program_Chairs · 2026-01-26

Reject